# Posters as a Tool to Improve Hand Hygiene among Health Science Students: Case—Control Study

**DOI:** 10.3390/ijerph182111123

**Published:** 2021-10-22

**Authors:** María Gázquez-López, Encarnación Martínez-García, Adelina Martín-Salvador, María Adelaida Álvarez-Serrano, Inmaculada García-García, Rafael A. Caparros-Gonzalez, María Ángeles Pérez-Morente

**Affiliations:** 1Faculty of Health Sciences, University of Granada, 51001 Ceuta, Spain; mgazquez@ugr.es; 2Faculty of Health Sciences, University of Granada, 18016 Granada, Spain; ademartin@ugr.es (A.M.-S.); igarcia@ugr.es (I.G.-G.); rcg477@ugr.es (R.A.C.-G.); 3Guadix High Resolution Hospital, Guadix, 18500 Granada, Spain; 4Instituto de Investigación Biosanitaria ibs. GRANADA, 18012 Granada, Spain; 5Faculty of Health Sciences, University of Jaén, 23071 Jaén, Spain; mmorente@ujaen.es

**Keywords:** hand hygiene, education, posters, nursing students, CFU colony-forming units

## Abstract

(1) Background: Numerous educational interventions have been conducted to improve hand hygiene (HH) compliance and effectiveness among nursing students, with mixed results. The aim is to evaluate the effectiveness of posters as a teaching tool and factors associated with HH quality. (2) Methods: A pre-post experimental intervention study was conducted with a total of 293 nursing students randomly assigned to two groups (experimental and control) who, before and after HH, took cell culture samples from their non-dominant hands. Only the experimental group was exposed to the poster. (3) Results: In the experimental group, significant differences were observed among students older than 22 years (*p* = 0.017; V = 0.188), with a higher percentage of failures (15.7% vs. 3.6%). Poster displaying was associated with passing, other variables being equal, although without statistical significance (ORa = 2.07; 95% CI = 0.81–5.26). Pre-practice hand contamination was weakly associated with lower HH quality (ORa = 0.99, 95% CI = 0.99–0.99). (4) Conclusions: The use of posters as a teaching method shows indications of efficacy. Prior hand contamination slightly affects the quality of HH. Further evaluation of teaching methods is needed to ensure good technical performance of HH to prevent the spread of infectious diseases during the COVID-19 pandemic.

## 1. Introduction

The effectiveness and utility of hand hygiene (HH) for infection prevention in healthcare is undisputed, and this is all the more true during the COVID-19 pandemic [1,2,3,4,5] While nurses generally tend to comply with HH recommendations [6], this form of hygiene is not always performed correctly [7,8,9,10].

Numerous interventions have been proposed to maintain adherence over time, with inconclusive results [11,12]. A number of interventions have been based on performance feedback or on placing alcohol-based hand rub (ABHR) at key points of care [11,13]. However, there is insufficient evidence to make specific recommendations on the content and implementation of such interventions [13].

Nursing students can act as potential vectors of infectious diseases during their clinical placements [7,8], which is why theory and practical training in HH is provided from the very beginning of their academic studies [7,14,15]. Overall, nursing students’ attitudes towards HH are favourable [7,10,16,17], but conventional teaching methods do not seem to provide a proper understanding of HH [7,18,19]. Additionally, Løyland et al. (2020) [20] confirmed that the adhesion to HH in the medical personnel, including nursing students, had a negative balance, which affected to the prevention of nosocomial infections and even prevented the reduction of the instructions of the antibiotic treatment in patients. In studies such as the one developed by Sundal et al. [19], the degree of general compliance in HH in nursing students was estimated to be 83.5% during their clinical internship. In these studies, the five moments of the HH of the WHO were evaluated: firstly, before touching the patient; secondly, before the cleaning/washing procedure; thirdly, after exposure to body fluids; fourthly, after touching the patient; and finally, after touching a patient in the environment [19,20]. Elola-Vicente et al. [21] evaluated the effectiveness of the HH technique in medical personnel. It would be advisable to consider it in the nursing students. A large body of research stresses the need to improve the training of future healthcare workers [8,10,18,22]. In addition to knowledge, other factors such as students’ sex, age, academic year, work experience, beliefs, perceived barriers, and attitudes influence the adoption of effective HH behaviours [7,8,23,24,25].

Innovative, multidisciplinary interventions have been proposed in the search for effective learning methods, with mixed results [7,9,14,15,18,26,27,28].

Posters have traditionally been used in health and social care as a resource to promote HH, albeit with poor results [29,30]. However, the WHO recommends the use of posters as reminders in the workplace and as tools for training healthcare workers [31]. To the best of our knowledge, there are no studies to date assessing the effectiveness of posters as a teaching method for improving the HH technique among nursing students.

This study has two aims: firstly, to evaluate the effectiveness of the poster as a tool for improving the quality of HH among nursing students; secondly, to determine the factors associated with correct HH among nursing students.

## 2. Materials and Methods

### 2.1. Study Design and Participants

An experimental pre–post intervention study was carried out at a public university in southern Spain during the 2019–2020 academic year. The nursing degree in Spain is divided into four years and students receive basic HH training in the first year, with clinical placements starting in the third year.

All undergraduate nursing students who had passed their basic HH training during the second, third, and fourth years and who voluntarily agreed to participate were invited to do so.

For a population of 321 students, the necessary sample size was estimated to be 140 participants, with a power of 95%, a 5% level of accuracy, and an expected proportion of 80%. Students were allocated to the intervention and control groups using random sampling stratified per academic year. To ensure that the practices ran smoothly, students were divided into groups of 20. The last group of fourth-year students, who were assigned to the control group, were not able to participate in the study due to the suspension of face-to-face tuition caused by the COVID-19 pandemic. However, the participation rate for that academic year was 74%. This study was performed according to the STROBE statement, Strengthening the Reporting of Observational studies in Epidemiology.

### 2.2. Description of the Intervention and Data Collection

The intervention consisted of two practice sessions per group led by the research team and an accredited laboratory technician. The first session lasted one hour and included a brief reminder of nosocomial infections, with special emphasis on the role of nursing professionals as the main vectors of transmission. The differences between the different HH techniques were defined, recalling the five moments recommended by the WHO, as well as the importance of keeping nails short, clean, and without nail polish or gel. Each student received two sterile swabs and a Petri dish with a previously identified agar culture. They then divided the plate into two equal parts to differentiate pre- and post-HH seeding and proceeded to sample and culture the non-dominant hand in the pre-HH part. Škodová et al. [25] confirmed that the most contaminated areas after HH are the thumbs and interdigital areas of the non-dominant hand. The decision was made to sample only that hand, in line with Cruz and Bashtawi [32], Elola-Vicente et al. [21], Silva et al. [33], and Škodová et al. [25].

The hands were then cleaned with ABHR. Sanitisers with alcohol concentrations above 60% have been shown to have similar efficacy to hand washing with soap and water. They were used because of their current availability in healthcare facilities [11,13]. All students were administered the same amount of ABHR with an alcohol concentration of 75% and performed the handwashing technique. The intervention group did so with the WHO poster displayed [34] and the control group did so without it. Finally, they performed a second sampling and seeding of the same hand in the post-HH zone. At the end of the process, the plates were placed in a culture oven at 35 °C for 48 h. HH and the Petri dish seeding process were supervised at all times by two members of the research team and an accredited laboratory technician.

This was followed by the second session, which lasted 15 min. Each student checked their plate and manually counted the colony forming units (CFU) under the supervision of their instructors. The results were recorded on a data sheet, which included socio-demographic and academic data. 

Figure 1 shows the group selection process. Figure 2 depicts the procedure for collecting and evaluating the results of the practice.

The variables to be considered were: group (intervention and control), age (continuous and dichotomised: ≤22 and >22), sex (male, female), academic year (second, third, fourth), conducting a clinical placement (yes, no), and pre- and post-HH contamination assessed using the number of CFU/cm^2^.

### 2.3. Data Analysis

HH effectiveness was assessed by classifying students as either pass or fail. The classification was conducted according to the recommendations of the Chinese Centre for Disease Control and Prevention, i.e., by counting the number of CFU/cm^2^. Aerobic bacterial counts are required to be under 10 CFU/cm^2^ among healthcare workers in general clinical units [35,36].

Descriptive statistics were applied using frequency and dispersion measures according to the nature of each variable. The Chi-square test was used to assess differences in age, gender, academic year, and year of study between the control and experimental groups. Pre- and post-intervention differences between and within groups were calculated using Spearman’s chi-square. The effect size was assessed using Cramér’s V and Cohen’s d statistic. Explanatory bivariate and multivariate logistic regression models were designed to adjust for all study variables, calculating crude and adjusted odds ratios, respectively, and their 95% CIs.

Every analysis has been conducted using the Statistical Package for the Social Sciences (SPSS) program, version 25, (IBM, New York, NY, USA, for Mac).

### 2.4. Ethical Considerations

This study was approved by the university centre and the Granada Research Ethics Committee (code number: 0100-N-21). Participants signed an informed consent form for data collection purposes in compliance with the European Directive 2001/20/EC and Spanish Law 14/2007 of 3 July on Biomedical Research.

## 3. Results

### 3.1. Characteristics of the Participants

After the CFU count, six plates (two belonging to the intervention group and four to the control group) were found to be contaminated and were therefore excluded. As a result, the final number of participants was 287. Their sociodemographic and academic characteristics are shown in Table 1.

### 3.2. Outcome of the Intervention

Table 2 shows the numbers and percentages of students in each group who were classified as either pass or fail before and after the intervention. A significant improvement was observed.

No differences in results were identified between the experimental group and the control group before and after performing HH with ABHR (Table 3).

Table 4 shows the distribution of sociodemographic and academic variables by study group after the workshops. In the experimental group, significant differences were observed by age (*p* = 0.017; *V* = 0.188), with students aged over 22 showing a higher percentage of fails (15.7%). A moderate association was found between pre-HH hand contamination and HH outcomes between the two groups (*p* = 0.005; *d* = 0.418), meaning that students who passed obtained a lower mean number of CFUs compared to those who failed.

The results of the bivariate and multivariate logistic regression are shown in Table 5. The use of posters as a teaching method for improving HH shows indications of effectiveness when adjusting for the other variables, although these are not statistically significant (OR = 2.07; 95% CI = 0.810–5.264). The number of CFUs prior to hand rubbing was slightly associated with the degree of cleanliness of the hands after the workshop (OR = 0.99; 95% CI = 0.991–0.999).

## 4. Discussion

This study explored the potential association between the use of a poster as a teaching tool and the quality of HH among nursing students in response to the need to find an effective method to improve the HH technique to control nosocomial infections among this group [25,32,37,38].

Our findings are in line with the results of numerous studies showing that the use of ABHR considerably reduces the microbial burden and is thus considered a suitable procedure for nosocomial infection control [25,27,32,39,40,41].

We observed no sex-based differences in the level of HH after using the poster as a teaching tool. Nonetheless, the available evidence on the influence of this variable based on various interventions remains controversial. Anderson et al. [42] and Pérez-Pérez et al. [38] point out that, regardless of the techniques used, women performed HH better than men in all cases. In contrast, Cruz and Bashtawi [32] report that being male and being in the first years of university study were predictive of greater knowledge of the technique. Recently, Merino-Plaza et al. [43] studied adherence to HH among healthcare professionals. Initially, men scored more poorly than women on adherence to HH. However, when targeted improvements in care services based on monitoring and feedback were implemented, men’s scores improved to match those of women. These strategies, which were also included in our training practices, may be behaving in a similar way in our study.

In the intervention group, we identified a relationship between age and the number of CFUs, whereby students aged 22 years and older had higher numbers of CFUs, perhaps as a result of being overconfident during the HH procedure [21,44]. Minervini [45] and Sancho [46] suggest that visual tools used for teaching, such as infographics, must be tailored to the characteristics of the students, including their age. Our findings suggest the need to adapt HH practices to take into account students’ ages.

Second-year students achieved the same results as those in senior years. This may be explained by the fact that knowledge of the subject increases in line with the academic year, resulting in an improvement in HH performance among senior students. However, Cruz and Bashtawi [32] failed to observe this relationship. Surprisingly, a number of authors argue that the closer undergraduates are to entering the labour market, the more confident they may feel about performing the technique, resulting in poorer performance [21,44]. According to Fernández-Prada et al. [37], the use of infographics as a teaching tool tends to improve the teaching–learning process during the first academic years. However, the overexposure to digital teaching materials that our students are currently experiencing may be diminishing the effect of posters with the passing of the academic years, leading to a shift away from the results reported by other researchers [47,48].

The multivariate model designed to explain the association between displaying the poster and the likelihood of passing has shown signs of effectiveness when adjusting for the other study variables, although it has not reached statistical significance. Visual tools certainly seem to be useful for optimising and accelerating comprehension processes and are often highly valued by students [29,47]. However, as Bicen and Behesti [49] point out, the main obstacle to the effectiveness of these tools is a lack of precise theoretical knowledge of the subject matter among students and excessive time required to analyse visual tools. The same may be said of infographics. Further studies are required to better assess the impact of these variables on the results obtained in our study.

Pre-HH hand contamination was slightly associated with a lower likelihood of passing. However, we agree with several authors in recommending that HH should be performed with soap and water whenever hands are visibly soiled or have been in contact with contaminants [34,50,51].

### Limitations

A potential classification bias may have been introduced by using the same Petri dish for pre- and post-HH culture. To avoid this limitation, we should have used one culture plate for each sample. However, only six plates were contaminated and excluded from the study. If this bias was present, it would be a non-differential bias across all comparison groups.

The last group of fourth-year students, who were assigned to the control group, were not able to participate in the study due to the suspension of face-to-face tuition caused by the COVID-19 pandemic. However, the participation rate for that academic year was 74%, which was satisfactory.

Another limitation could be the presence of an observer during practice. This could result in student nurses washing their hands more thoroughly than usual and obtaining better results regardless of the poster. However, it would be a non-differential bias, since it would affect the two comparison groups equally.

## 5. Conclusions

This practice-based teaching method combining HH with ABHR and the display of a specific poster on HH shows indications of being effective in improving the quality of HH among nursing students. However, further research is required to confirm this association. A high level of hand contamination prior to the practices decreased the likelihood of achieving a good level of cleanliness. Further evaluation of teaching methods to ensure good technical performance of HH at university level is required to prevent the spread of infectious diseases during the COVID-19 pandemic.

The lack of a constant presence of an observer in future work in the hospital may significantly affect the frequency and implementation of hand-washing procedures.

The evaluation of new HH teaching methods in the college setting ensures that future nurses are educated and sensitised to the spread of infectious diseases during the COVID-19 pandemic.

Improving the hand hygiene procedure is possible if the level of hygiene awareness increases and future health care workers are convinced of the legitimacy and necessity of the effective application of hygiene procedures (internal motivation).

## Figures and Tables

**Figure 1 ijerph-18-11123-f001:**
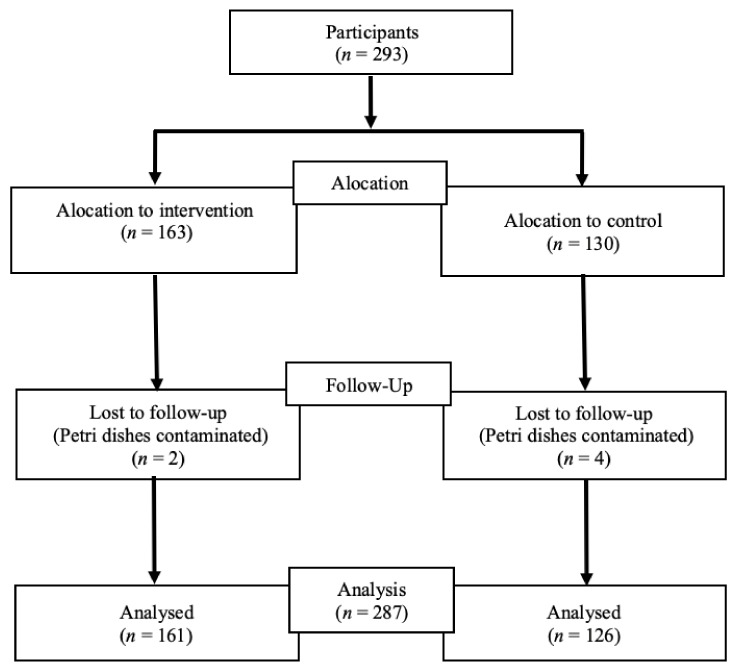
The group selection process.

**Figure 2 ijerph-18-11123-f002:**
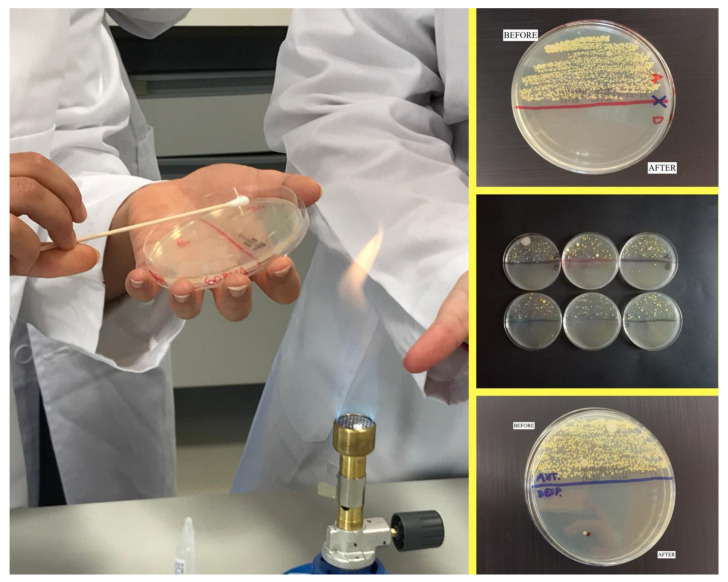
Description of the intervention and data collection.

**Table 1 ijerph-18-11123-t001:** Characteristics of the participants.

	Participants	Control Group	Intervention Group	
	(*n* = 287)	(*n* = 126)	(*n* = 161)	
	*M* (SD)	*M* (SD)	*M* (SD)	*p*
Age (years)	22.5 (4.43)	22 (2.81)	22.89 (5.34)	0.686
	*n* (%)	*n* (%)	*n* (%)	
Age (dichotomised)				
≤22	185 (64.5)	83 (65.9)	102 (63.4)	0.658
>22	102 (35.5)	43 (34.1)	59 (36.6)	
Sex				
Male	119 (41.5)	59 (46.8)	60 (37.3)	0.103
Female	168 (58.5)	67 (53.2)	101 (62.7)	
Academic year				
Second	110 (38.3)	52 (41.3)	58 (36)	0.660
Third	95 (33.3)	40 (31.7)	55 (34.2)	
Fourth	82 (28.2)	34 (27)	48 (29.8)	
Conducting a clinical placement				
No	110 (38.3)	52 (41.5)	58 (36.5)	0.364
Yes	177 (61.7)	74 (58.7)	103 (64.1)	

Values are shown as means (standard deviations) or frequencies (percentages).

**Table 2 ijerph-18-11123-t002:** Intervention data (HH).

	Participants (*n* = 287)			Control Group (*n* = 126)			Intervention Group (*n* = 161)		
	Pre-HH *n* (%)	Post-HH *n* (%)	*p*	Pre-HH *n* (%)	Post-HH *n* (%)	*p*	Pre-HH *n* (%)	Post-HH *n* (%)	*p*
Pass	24 (8.4)	262 (91.3)	0.001 **	8 (6.3)	119 (94.4)	0.001 **	16 (9.9)	143 (88.8)	0.001 **
Fail	263 (91.6)	25 (8.7)		118 (93.7)	7 (5.6)		145 (90.1)	18 (11.2)	

*p* compares the pre- and post-test in the control group, intervention group, and total group. ** *p* < 0.01.

**Table 3 ijerph-18-11123-t003:** Intervention data (HH). Independent sample results.

	Control Group (*n* = 126)	Intervention Group (*n* = 161)		Control Group (*n* = 126)	Intervention Group (*n* = 161)	
	Pre-HH *n* (%)	Pre-HH *n* (%)	*p*	Post-HH *n* (%)	Post-HH *n* (%)	*p*
Pass	8 (6.3)	16 (9.9)	0.276	119 (94.4)	143 (88.8)	0.094
Fail	118 (93.7)	145 (90.1)		7 (5.6)	18 (11.2)	

Pre-test *p* compares the pre-test between the control group and the intervention group. Post-test *p* compares the post-test between the control group and the intervention group.

**Table 4 ijerph-18-11123-t004:** Post-HH data: comparisons based on sociodemographic and academic variables.

	Control Group (*n* = 126)			Experimental Group (*n* = 161)			
	Post-HH			Post-HH			
	Pass *n* (%)	Fail *n* (%)	*p*	Pass *n* (%)	Fail *n* (%)	*p*	*V*
Sex							
Male	57 (96.6)	2 (3.4)	0.319	50 (83.3)	10 (16.7)	0.089	
Female	62 (92.5)	5 (7.5)		93 (92.1)	8 (7.9)		
Age							
≤22	39 (90.7)	4 (9.3)	0.186	57 (96.6)	2 (3.4)	0.017 *	0.188
>22	80 (96.4)	3 (3.6)		86 (84.3)	16 (15.7)		
Academic year							
Second	49 (94.2)	3 (5.8)	0.983	52 (89.7)	6 (10.3)	0.589	
Third	38 (95)	2 (5)		47 (85.5)	8 (14.5)		
Fourth	32 (94.1)	2 (5.9)		44 (91.7)	4 (8.3)		
Conducting a clinical placement							
No	70 (94.6)	4 (5.4)	0.930	91 (88.3)	12 (11.7)	0.801	
Yes	49 (94.2)	3 (5.8)		52 (89.7)	6 (10.3)		
	*M* (SD)	*M* (SD)		*M* (SD)	*M* (SD)		*d*
Pre-HH hand contamination	84.29 (77.79)	112.57 (44.56)	0.344	91.18 (85.30)	156.56 (131.02)	0.005 **	0.418

*p* compares the pre- and post-test in the control group and intervention group. * *p* < 0.05. ** *p* < 0.01. (*V*) = Cramér’s *V*. (*d*) = Cohen’s *d.*

**Table 5 ijerph-18-11123-t005:** Logistic regression for students who passed (post-HH).

	ORc	95% CI	ORa	95% CI
Poster				
No	1		1	
Yes	0.47	(0.189–1.157)	2.07	(0.810–5.264)
Sex				
Male	1		1	
Female	1.34	(0.588–3.043)	0.69	(0.282–1.679)
Age				
≤22	1		1	
>22	0.18	(0.707–4.743)	0.50	(0.188–1.328)
Conducting a clinical placement				
No	1		1	
Yes	0.90	(0.382–2.106)	1.18	(0.464–3.022)
Pre-HH hand contamination	0.99	(0.991–0.998)	0.99	(0.991–0.999)

Hosmer–Lemeshow goodness-of-fit test: χ^2^ (8) = 8.42; *p* = 0.394. OR: Odds Ratio; CI: Confidence Interval.

## Data Availability

The data that support the findings of this study are available from the corresponding author, upon reasonable request.

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
