# Peer review of "Posters as a Tool to Improve Hand Hygiene among Health Science Students: Case—Control Study"

_ijerph, 2021, doi:10.3390/ijerph182111123_

Round 1
Reviewer 1 Report
Dear Authors,
The manuscript (ijerph-1427832) presented for review is very interesting, and after minor correction and completing the missing information, I recommend it for publication.
This study is very important as there is a need to improve the quality of handwashing by healthcare professionals. Therefore, it is justified to look for methods to improve hand hygiene.
Authors, Please note and address the following comments:
Material and methods
I have a few questions about data collection. Who controlled the washing of hands and seeding of the Petri dishes? If I correctly understood data collection, the students themselves seeded Petri dishes?
There is a lack of detailed information on how the Petri dishes were prepared for data collection.
2.4. Measurement tools – Where is the data collected by the authors, and their collection is described in this chapter.
Conclusion
Could the presence of an observer have influenced the obtained results? If yes, the authors should add information on this in the Conclusions section. The lack of a constant presence of an observer in future work in the hospital may significantly affect the frequency and implementation of hand-washing procedures. I propose to the authors to add another conclusion. Improving the hand hygiene procedure is possible if the level of hygiene awareness increases and future health care workers are convinced of the legitimacy and necessity of the legitimacy application of hygiene procedures (internal motivation).
Limitations
As a limitation of the study, the authors indicated the use of the same Petro dish for pre-and post-HH culture. I reckon the second limitation is the presence of an observer during hand washing by nursing students. Probably that students washed their hands more thoroughly than usual when someone was watching them.
References
Authors are requested to check the correctness of citation of the references by the requirements of the International Journal of Environmental Research and Public Health. In my opinion, now it is not correct. The year of publication should be bold. The name of the journal should be written in italics. There is no DOI number.
Technical notes
Figure 1: The title of this drawing is repeated below and above the Figure.
Despite my comments, I am pleased to recommend this manuscript for publication.
Best of luck with your paper and be safe!
Reviewer
Reviewer 2 Report
First of all I want to thank the opportunity to review this article. Below I will make some considerations that I hope will help to improve the article.
-Introduction:
-You say (line 40):"Additionally, Løyland, Peveri, Hes, Sevaagbakke, Taasen & Lindeflaten (2020)". In this case it would not be necessary to cite all the authors. It's correct to put the first author and et al.
-"In these studies, the 5 moments of the HH of the WHO were evaluated". In this sentence I miss a reference, or an explanation of these 5 phases.
Results:
-It would be conveninet to homogenize the presentation of the tables. For example in Table 1, the units (n, %..) are placed at the end of the table and instead in Table 2 they are presented in an integrated way.
-I think it would be necessary to better clarify the results related to the attitudes of the students. While it is explained very well in the methods section, there are no clear results related to this variable
-Note that in Table 3 the word intervention is separated.
